# Carbene-catalyzed atroposelective synthesis of axially chiral styrenes

Jia-Lei Yan [1,4], Rakesh Maiti[1,4], Shi-Chao Ren[1], Weiyi Tian [2✉], Tingting Li[3], Jun Xu [1,2], Bivas Mondal[1], Zhichao Jin[3] & Yonggui Robin Chi [1,3✉]

Axially chiral styrenes bearing a chiral axis between a sterically non-congested acyclic alkene and an aryl ring are difficult to prepare due to low rotational barrier of the axis. Disclosed here is an *N*-heterocyclic carbene (NHC) catalytic asymmetric solution to this problem. Our reaction involves ynals, sulfinic acids, and phenols as the substrates with an NHC as the catalyst. Key steps involve selective 1,4-addition of sulfinic anion to acetylenic acylazolium intermediate and sequential *E*-selective protonation to set up the chiral axis. Our reaction affords axially chiral styrenes bearing a chiral axis as the product with up to > 99:1 *e.r.*, > 20:1 *E/Z* selectivity, and excellent yields. The sulfone and carboxylic ester moieties in our styrene products are common moieties in bioactive molecules and asymmetric catalysis.

[1] Division of Chemistry & Biological Chemistry, School of Physical & Mathematical Sciences, Nanyang Technological University, Singapore 637371, Singapore. [2] Guizhou University of Traditional Chinese Medicine, Guiyang 550025, China. [3] Laboratory Breeding Base of Green Pesticide and Agricultural Bioengineering, Key Laboratory of Green Pesticide and Agricultural Bioengineering, Ministry of Education, Guizhou University, Huaxi District, Guiyang 550025, China. [4]These authors contributed equally: Jia-Lei Yan, Rakesh Maiti. ✉email: tianweiyi@gzy.edu.cn; robinchi@ntu.edu.sg

Axially chiral molecules are widely used as catalysts and ligands in asymmetric catalysis[1–3]. This class of natural or synthetic molecules has also shown an increasing presence in bioactive molecules for medical[4–7] and agricultural[8,9] uses. Among the well-known axially chiral scaffolds, most of the chiral axis is between two aromatic moieties. Many synthetic and preparation methods are now available for relatively efficient and scalable access to this class of molecules such as axially chiral biaryls[10–14]. In addition, axially chiral molecules such as benzamides[15–17] and anilides[18–22] have also received considerable attention. In contrast, axially chiral styrenes, which bear a chiral axis between a simple alkene and an aryl ring, are much less developed although this kind of chirality was realized and intensively studied by Adams and co-workers in the 1940s[14,23,24]. Atroposelective access to axially chiral styrenes, especially those bearing acyclic alkene units, is challenging due to low rotational barrier, weak configurational stability, and difficult control of the alkene $E/Z$ selectivity (Fig. 1a)[23,24]. In the past, synthetic success was mainly limited to aryl-cycloalkene connections in which the alkene is trapped in a ring to imitate the rigidity of biaryls and increase conformational stability[25–32]. It is only in recent years that several approaches emerged for asymmetric access to axially chiral aryl-acyclic alkene scaffolds. These remarkable studies include chiral amines catalyzed addition of carbon anions to ynals[33], Brønsted acids catalyzed addition of nucleophiles (e.g., sulfinic anions, 5H-oxazol-4-ones, and electrorich aromatic units) to in situ generated allenes[34–43], transition metal-catalyzed cross coupling[44–46], and desymmetrization/kinetic resolution strategies (Fig. 1b)[47–51]. Despite this progress, it is still highly demanded to develop powerful atroposelective strategies for rapid access to axially chiral styrenes bearing acyclic alkenes.

Here, we disclose an $N$-heterocyclic carbene (NHC)-catalytic approach for highly atroposelective and efficient synthesis of axially chiral styrenes bearing both sulfone and carboxylic ester units (Fig. 1c). In the confined arena of axially chiral molecule synthesis, NHC catalysis was previously used by us[52,53] and others[30,54–64] to mediate cycloaddition[30,52,54–58], desymmetrization[53,59–61] or kinetic resolution[62–64] to construct molecules bearing the chiral axis between two rigid rings. In our present study, nucleophilic 1,4-addition of sulfur atom of sulfinic acid to an NHC-activated ynal set up the chiral axis. This nucleophilic addition is a chemoselective process as another nucleophile (phenol) and an electronic reactive site (the carbonyl carbon of the acyl azolium moiety) are simultaneously present in the same reaction system. The chiral axis is well controlled with up to >99:1 $e.r.$, and the alkene formation is also highly selective with the exclusive formation of the $E$-alkene moiety in most of the cases. Both the sulfone and carboxylic ester groups in our axially chiral styrene products are common structural motifs in bioactive and other functional molecules[65,66]. This methodology opened a new gate for accessing axially chiral styrenes bearing acyclic alkenes.

## Results

**Optimization of reaction conditions**. We initiated our studies by using ynal **1a** and sulfinic acid **2a** as model substrates with 3,3′,5,5′-tetra-$tert$-butyldiphenylquinone (DQ) as an oxidant to search for suitable conditions. Preliminary results revealed that sodium acetate and diethyl ether are suitable base and solvent, respectively (Supplementary Tables 1–3). Further optimization of conditions is summarized in Table 1. The use of aminoindanol-derived precatalyst with an $N$-mesityl substituent (ent-**A**) afforded the desired product with moderate yield (54%) and enantioselectivity (22:78 $e.r.$, entry 1). Switching the $N$-mesityl unit of catalyst **A** to the electron-rich 2,6-dimethoxyphenyl group (to get catalyst **B**) did not improve the enantioselectivity (entry 2). When

NHC precatalyst **C** with an electron-deficient trichlorophenyl group was used, no desired product formed (entry 3). Examination of the steric hindrance of the $N$-substitutes on NHC catalyst showed that the introduction of a bulkier $N$-triisopropylphenyl (**D**) or $N$-tricyclohexylphenyl (**E**) substituent resulted in improved enantioselectivities (entries 4–5). We next chose NHC precatalyst **E** for further optimization and found that reducing the reaction temperature to 0 °C gave an apparent improvement in enantioselectivity (91:9 $e.r.$, entry 6). However, the yield of the product was still moderate due to the competitive direct 1,2-addition of isopropanol to acetylenic acylazolium intermediates (**II**). Therefore, we further screened several nucleophiles to improve the results (entries 7–12) and delightfully found a sharp increase in yield when phenols were used (entries 8–12). Among the phenols, 2-methoxyphenol showed the best performance in the reaction, affording product **9a** in 97% yield with >99:1 $e.r.$ and $E/Z$ 20:1 (entry 12). Further improvement in the $E/Z$ value was observed by reducing the reaction temperature to −20 °C. The catalyst loading could also be reduced to 5 mol%, leading to the desired product (**9a**) in 94% yield with > 99:1 $e.r.$ and $E/Z$ > 20:1 (entry 13).

**Substrate scope**. With the optimal reaction conditions in hand (Table 1, entry 13), we next explored the generality of the reaction. Initially, we examined the scope of sulfinic acids (**2**) by using **1a** as model substrate (Fig. 2a). Both electron-donating ($CH_3$, $OCH_3$) and electron-withdrawing ($CF_3$, CN, halogens) groups were well tolerated, providing the desired axially chiral styrenes (**9b–9i**) in good to excellent yields with high optical purities and $E/Z$ ratios. The steric effect of the substituents on the benzene ring has a considerable influence on the reaction outcome. For example, introducing a methyl group at the $para$-position of the benzene ring gave the desired product **9b** in excellent results. However, when the methyl group was installed at the $ortho$-position, the corresponding product (**9c**) was obtained in moderate yield (45%) with lower enantioselectivity (82.5:17.5 $e.r.$), which probably due to the steric repulsion between the methyl group and acetylenic acylazolium intermediate (**II**). The phenyl group of sulfinic acids could be replaced with heteroaryl group (e.g., thiophenyl) without affection on the reaction efficiency, affording **9j** in 99% yield with 98.5:1.5 $e.r.$ and $E/Z$ > 20:1. Moreover, aliphatic sulfinic acids were also compatible in our catalytic system providing the desired axially chiral styrenes (**9k–9l**) in good yields with high $e.r.$ values, albeit with lower $E/Z$ ratios than aromatic sulfinic acids.

The scope of ynals (**1**) was also investigated by using benzenesulfinic acid (**2a**) as the model substrate (Fig. 2b). The steric and electronic effects on the naphthalene ring of ynals were evaluated by tuning the substitution patterns. A wide range of electron-donating (Et, $OCH_3$, OEt, OBn, OMOM, etc.) and electron-withdrawing ($CO_2CH_3$, CN, halogens) substituents at different positions (2-, 3-, 6-, 7-) of naphthalene ring were well tolerated in our catalytic reaction, giving the corresponding axially chiral styrenes in good to excellent yields with high enantioselectivities and $E/Z$ ratios in most of the cases. Notably, 2-alkoxyl substituents on naphthalene ring generally gave better results compared to other substituents, such as 2-alkyl and ester substituents (**9m, 9q–9r**). Inspired by the coordination effect of Lewis acid with nucleophiles and 2-methoxyl substituted acetylenic acylazolium intermediates[30], we hypothesized that there might be hydrogen bond interactions between nucleophiles and 2-alkoxyl substituents in the catalytic process, which probably had a positive effect on the enantio- and $E/Z$ selectivity (Supplementary Fig. 190). Besides, other possibilities (e.g., resonance intermediates pathway; Supplementary Fig. 191) might also exist to explain the reaction outcome. Bulkier groups on the

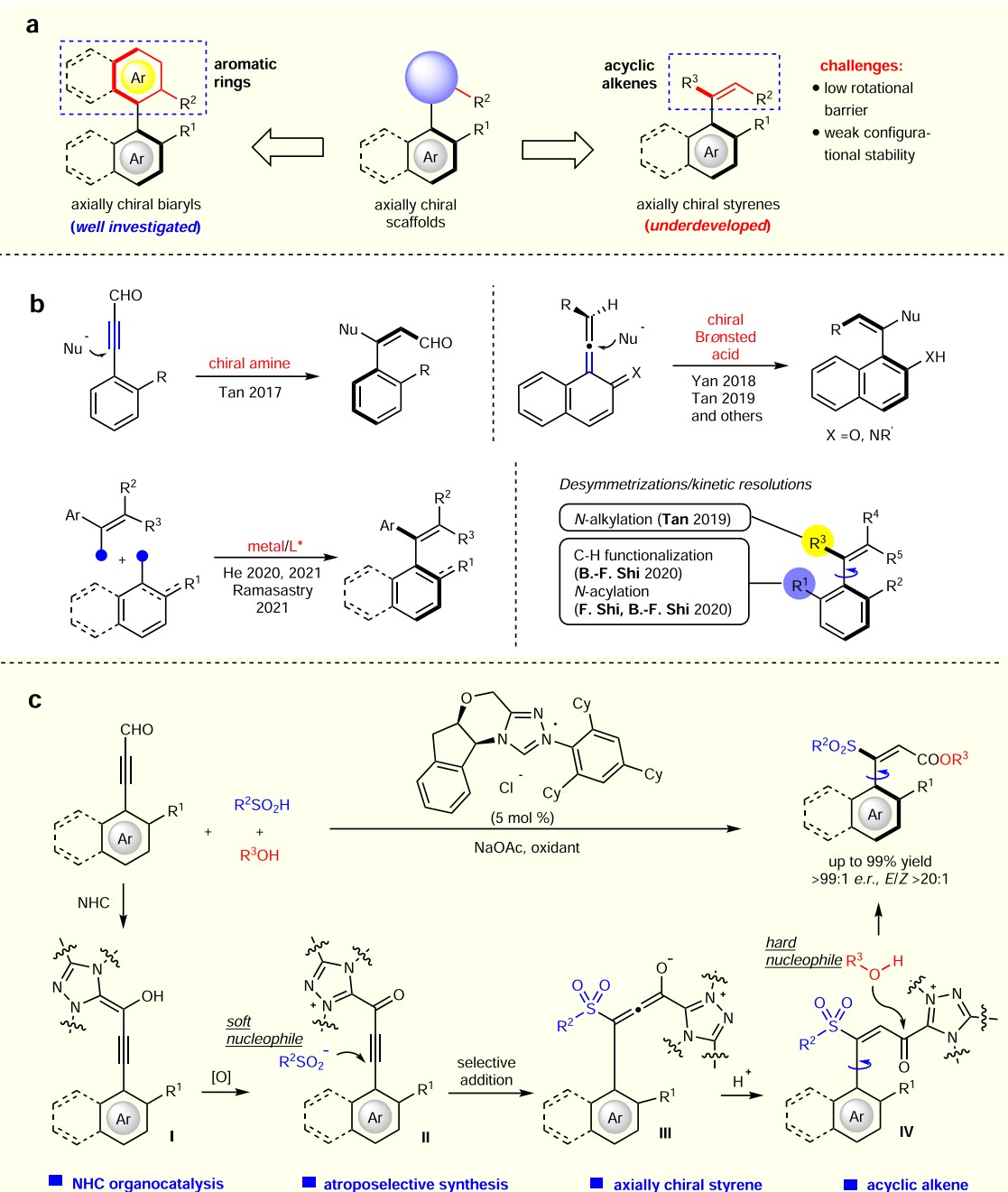

**Fig. 1 Challenges and strategies to access axially chiral styrenes bearing acyclic alkenes. a** Challenges in the synthesis of axially chiral styrenes bearing acyclic alkenes. **b** Previous methods for accessing axially chiral styrenes bearing acyclic alkenes. **c** Our Strategy: carbene-catalyzed chemo- and enantio-selective addition.

2-position of naphthalene ring (**9r**) and strong electron-withdrawing groups (**9q**, **9y–9z**) also had a negative effect on both yield and enantioselectivity. To our delight, the β-naphthyl unit of ynal (**1**) can be replaced with heteroaryl substituents, such as quinoline (**9ac**), indole (**9ad**) and benzothiophene (**9ae**). In these cases, moderate to good yields and high *e.r.* values were regularly obtained, although the *E/Z* selectivity decreased for **9ae**. When the naphthalene ring in ynals was replaced with phenyl ring, high enantioselectivities could still be achieved, but the *E/Z* ratios dropped sharply (**9af–9ag**). The low *E/Z* selectivities for **9ae–9ag** might attribute to the less steric hindrance of aromatic rings on the ynal skeletons, which may lead to more flexibility of the allenolate intermediates (**III**) and thus cause lower selectivity of the *E*-protonation (Supplementary Fig. 192).

**Stability evaluation for the axially chiral products**. The stereochemical stabilities of our axially chiral styrenes were evaluated via both experimental and computational methods. We monitored the *ee* value of **9a** over 24 h at different temperatures in toluene, and found **9a** racemized quickly at 100 °C, but much slower at 75 °C and 50 °C (Supplementary Fig. 196). Notably, a 2-month-term monitor showed no obvious *ee* drop of **9a** at room temperature. The experiment-based calculation (Supplementary Fig. 197) revealed that the rotational barrier of **9a** is $\Delta G^{\ddagger} = 30.3$ kcal mol$^{-1}$ (Fig. 3), which is in accordance with the value ($\Delta G^{\ddagger} = 30.1$ kcal mol$^{-1}$) resulting from the density functional theory (DFT) calculation. The computationally derived energy scan for **9a** over dihedral angles from −180° to 180° also

**Table 1 Optimization of the reaction conditions[a].**

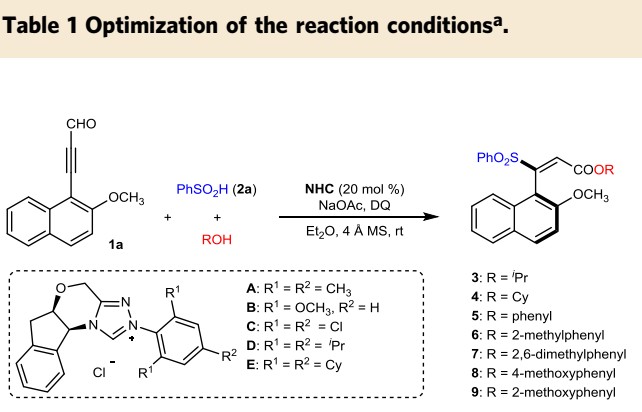

| A: R¹ = R² = CH₃ |
|---|

A: R¹ = R² = CH₃
B: R¹ = OCH₃, R² = H
C: R¹ = R² = Cl
D: R¹ = R² = ⁱPr
E: R¹ = R² = Cy

3: R = ⁱPr
4: R = Cy
5: R = phenyl
6: R = 2-methylphenyl
7: R = 2,6-dimethylphenyl
8: R = 4-methoxyphenyl
9: R = 2-methoxyphenyl

| Entry | NHC | ROH | Product (yield)[b] | e.r.[d] | E/Z[e] |
|---|---|---|---|---|---|
| 1 | ent-**A** | ⁱPrOH | **3** (54%)[c] | 22:78 | >20:1 |
| 2 | **B** | ⁱPrOH | **3** (40%)[c] | 78:22 | >20:1 |
| 3 | **C** | ⁱPrOH | **3** (0%)[c] | – | – |
| 4 | **D** | ⁱPrOH | **3** (37%)[c] | 85:15 | >20:1 |
| 5 | **E** | ⁱPrOH | **3** (40%)[c] | 87:13 | >20:1 |
| 6 | **E** | ⁱPrOH | **3** (48%)[c] | 91:9 | >20:1 |
| 7 | **E** | cyclohexanol | **4** (trace) | – | – |
| 8 | **E** | phenol | **5** (90%) | 86:14 | 20:1 |
| 9 | **E** | 2-methylphenol | **6** (91%) | 83:17 | >20:1 |
| 10 | **E** | 2,6-dimethylphenol | **7** (65%) | 84:16 | >20:1 |
| 11 | **E** | 4-methoxyphenol | **8** (97%) | >99:1 | 15:1 |
| 12 | **E** | 2-methoxyphenol | **9a** (97%) | >99:1 | 20:1 |
| 13[f] | **E** | 2-methoxyphenol | **9a** (94%) | >99:1 | >20:1 |

[a]Reaction conditions: **1a** (0.05 mmol), **2a** (0.1 mmol), NHC precatalyst (20 mol%), NaOAc (0.15 mmol), DQ (0.1 mmol), ROH (0.055 mmol), Et₂O (1 mL), and 4 Å MS (100 mg); for entries 1–5, rt,12 h; for entries 6–12, 0 °C, 36 h.
[b]Isolated yield unless otherwise noted.
[c]NMR yield with 1,1,2,2-tetrachloroethane as an internal standard.
[d]e.r. = the ratio of enantiomers, determined via HPLC on a chiral stationary phase.
[e]E/Z = the ratio of E and Z isomers, determined via ¹H NMR analysis of the crude reaction mixture.
[f]**E** (5 mol%), −20 °C, 6 d. 4 Å MS = 4 Å molecular sieves.

reagent in biorthogonal click chemistry[67] to form triazole linkages such as **14**. In addition, our products could also be converted to oxazoline-containing axially chiral molecules (e.g., **13**) which could be potentially used as ligands in asymmetric catalysis. The halogen groups on our axially chiral products give opportunities for further functionalization via transition metal-catalyzed cross-coupling reactions. For example, the Sonogashira coupling[68] between **9x** and phenylacetylene furnished **15** in 82% yield with >99:1 e.r. and E/Z > 20:1. Further transformations based on sulfone unit[69] are also promising although several trials in our study were not successful. Actually, the sulfone unit in our products is a common moiety in pharmaceutically relevant compounds[65,66,70], which indicates the potential utility of our methodology in pharmaceutical synthesis.

## Discussion

In summary, we have developed an NHC catalytic strategy for atroposelective synthesis of the challenging axially chiral styrenes bearing acyclic alkenes. Our reaction involves ynals, sulfinic acids, and phenols as the substrates. Mild reaction conditions are used to provide a broad scope of products with moderate to high yields (up to 99%) and excellent stereoselectivities (up to >99:1 e.r., E/Z > 20:1). We have also evaluated the stabilities of the axially chiral styrenes via both experimental and computational methods and found that these molecules are quite stable at relatively low temperatures. Both the sulfone and carboxylic ester groups in our axially chiral styrene products are common structural motifs in bioactive and other functional molecules, indicating the potential utility of our methodology in pharmaceutical synthesis. Our products bear functional groups that can be readily transformed to diverse axially chiral molecules, although there is a limit that the sulfone unit was not successfully functionalized in our study.

Despite that only sulfinic acids were employed to furnish the chemo- and enantio-selective 1,4-addition in the present research, other nucleophiles (e.g., phosphines, enol relevents, and electron-rich aromatic units) are also potentially suitable in our catalytic system as related studies in our laboratories have shown encouraging results. Thus we believe our methodology would inspire a further study of axially chiral styrenes especially those bearing acyclic alkenes. Further studies on atroposelective synthesis of axially chiral molecules, especially those with potential applications in medicines and agricultural chemicals, are ongoing in our laboratories.

## Methods

**General information**. All reactions were conducted in oven-dried glassware under an atmosphere of dry nitrogen or argon. All reaction solvents were purified before use: Tetrahydrofuran, diethyl ether was distilled from Na/benzophenone. Dichloromethane, dimethylformamide, triethylamine were distilled from CaH₂. The procedures for preparation of NHC pre-catalysts, sulfinic acids, and ynals are provided in supplementary materials ¹H NMR and ¹³C NMR spectra were recorded with Bruker spectrometers using deuteriochloroform as solvent. High-resolution mass spectra (HRMS) were obtained on Finnigan MAT 95 XP mass spectrometer (Thermo Electron Corporation) and are reported as m/z (relative intensity). Optical rotations were measured using a 1 mL cell with a 1 dm path length on a Jasco P-1030 polarimeter. All the e.r. values were determined via chiral high-performance liquid chromatography (HPLC) analysis using Shimadzu LC-20AD HPLC workstation. All the E/Z ratios were determined via ¹H NMR analysis of the crude reaction mixture.

**General procedure for the synthesis of axially chiral styrenes**. To a 10 mL oven-dried Schlenk tube equipped with a magnetic stir bar, was added the aldehyde (0.1 mmol), NHC pre-catalyst **E** (2.9 mg, 0.005 mmol), 4 Å molecular sieves (150 mg), sulfinic acid (28.4 mg, 0.2 mmol), sodium acetate (24.6 mg, 0.3 mmol), 3,3′,5,5′-tetra-tert-butyldiphenylquinone (81.8 mg, 0.2 mmol). The Schlenk tube was sealed with a septum, evacuated, and refilled with nitrogen (3 cycles). 2-Methoxyphenol (12 μL, 0.11 mmol) and diethyl ether (2 mL) were then added via syringe. The Schlenk tube was quickly transferred to a −20 °C chiller, and the reaction mixture was stirred at −20 °C for 6 days. After completion of the reaction, monitored by TLC plate, the reaction mixture was concentrated in vacuo and the

indicated the rotational barrier of **9a** is about 30 kcal mol⁻¹ (Supplementary Fig. 193). Similarly, the rotational barriers for **9n** and **9o** were also measured experimentally at 100 °C in toluene (**9n**, $\Delta G^\ddagger = 31.0$ kcal mol⁻¹, $t_{1/2rac} = 18.1$ h; **9o**, $\Delta G^\ddagger = 31.2$ kcal mol⁻¹, $t_{1/2rac} = 25.1$ h). From these results, we can clearly see that our axially chiral products are stable at relatively low temperatures, and the products bearing a bulker 2-substituent on the naphthalene ring would generally be more stable than that bearing a less bulky 2-substituent.

**Scale-up synthesis and versatile transformations**. Our NHC catalytic approach is amenable to gram-scale synthesis, with the formation of axially chiral styrene **9a** (1.24 g) in 87% yield with 98:2 e.r. and E/Z > 20:1. The optically enriched product could be further functionalized through simple protocols (Fig. 4). For example, our axially chiral styrenes were suitable for ester 1,2-addition reactions with Grignard reagents (**10a**–**10b**). The afforded alcohol (**10a**) could be further functionalized (e.g., silylation and alkylation, see the Supplementary Information for details) without loss of e.r. and E/Z value. The ester unit of **9a** could undergo a transesterification or aminolysis to afford different kinds of ester-containing styrenes (**11a**–**11c**), or amide-containing styrenes (**12**) in high yields without erosion of optical purity. It is worthy to note that even the bulky L-menthol derived ester (**11c**) could be afforded efficiently as well. The alkyne-containing product (**11b**) can be used as a coupling

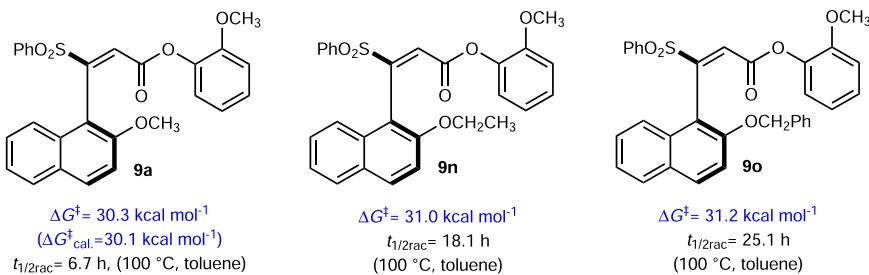

**Fig. 2 Scope of the reaction. a** Variations of sulfinic acids. **b** Variations of ynals. Standard reaction conditions: **1a** (0.1 mmol), **2a** (0.2 mmol), **E** (5 mol%), NaOAc (0.3 mmol), DQ (0.2 mmol), 2-methoxyphenol (0.11 mmol), Et$_2$O (2 mL), 4 Å MS (150 mg), −20 °C, 6 d. $^a$**E** (20 mol%) was used at −40 °C for 7 d. $^b$The reaction was performed at −20 °C for 9 d. $^c$**E** (20 mol%) was used.

**Fig. 3 The rotational barriers (△$G^‡$) and half-lives ($t_{1/2rac}$) of 9a, 9n–9o.** The △$G^‡$ values were obtained via racemization experiment. The △$G^‡_{cal.}$ value was obtained via DFT calculation. The half-lives ($t_{1/2rac}$) were determined via analysis of experimental data.

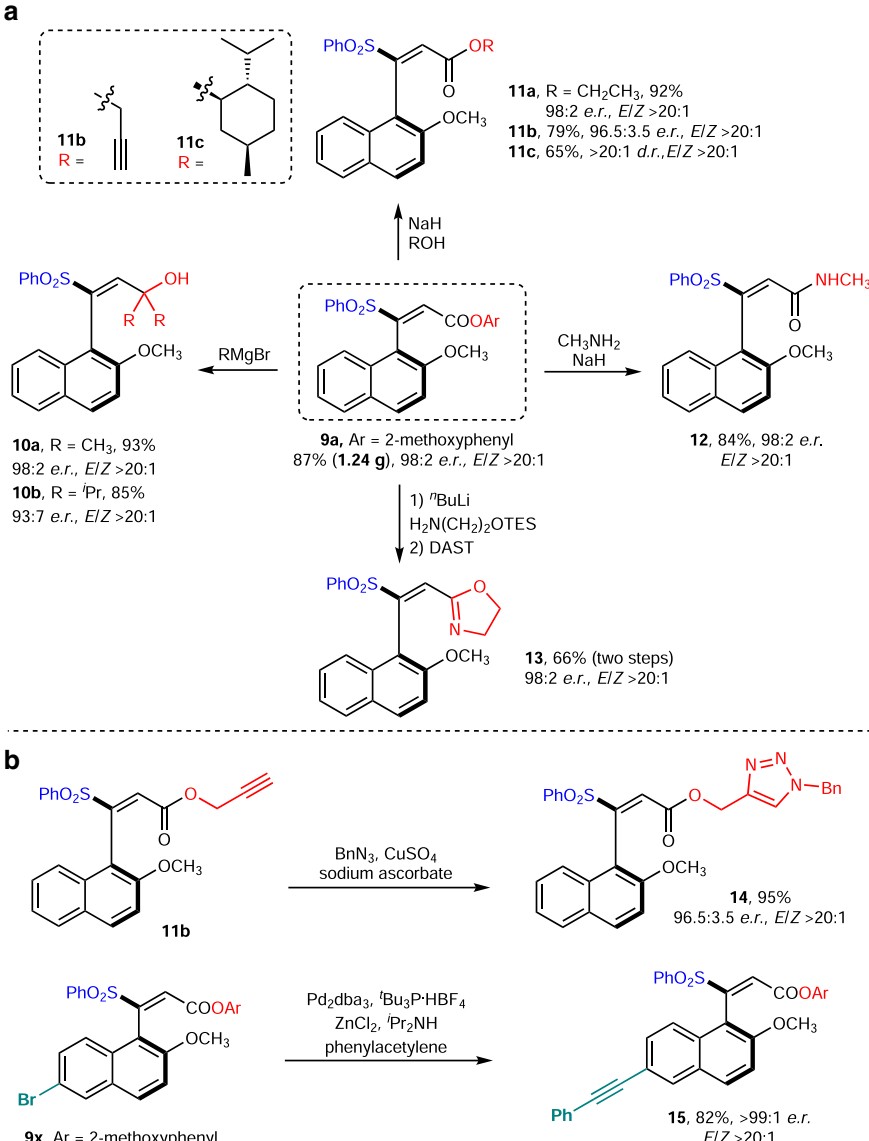

**Fig. 4 Versatile chemical transformations of the axially chiral products. a** Synthetic transformations of the axially chiral product (**9a**) based on ester group. **b** Synthetic transformations of **11b** and **9x**.

residue was subjected to column chromatography directly using ethyl acetate/hexanes as eluent to afford the desired product.

The gram-scale synthesis of **9a** was performed according to the general procedure using **1a** (630.7 mg, 3.0 mmol), **2a** (853.0 mg, 6.0 mmol), NHC pre-catalyst **E** (86.0 mg, 0.15 mmol), sodium acetate (738.0 mg, 9.0 mmol), 3,3′,5,5′-tetra-*tert*-butyldiphenylquinone (2.45 g, 6.0 mmol), 2-methoxyphenol (363.0 μL, 3.3 mmol) 4 Å molecular sieves (3.0 g) and diethyl ether (60 mL). After stirring at −20 °C for 6 days and the subsequent column chromatography purification, the product **9a** (1.24 g, 87% yield, 98:2 *e.r.*, E/Z > 20:1) was afforded.

**DFT calculations on the rotation barrier**. The energetic barriers of the racemization process of **9a** were investigated computationally by the DFT calculation. All calculations were carried out using the Gaussian 16 C.01 program package. The geometry optimizations were performed using a hybrid B3LYP exchange-correlation. The 6–31 G (d, p) basis set was used for C, H, O, and S atoms. Vibrational frequency calculations were performed (at 303.15 K) to characterize the nature of each stationary point. A tight convergence ($10^{-12}$ au) criterion was employed, and the solvent toluene ($\varepsilon = 2.40$) was considered using the SMD continuum solvent model (UFF radii). Single-point calculations were carried out for each optimized structure by using M06-2X functional and Def2-TZVP basis sets. The ground-stage structures (**9a**, *ent*-**9a**) and a transition state (**TS9a**) corresponding to the rotation along two directions were located. The rotational barrier

($\Delta G^{\ddagger}$) for enantiomerization was obtained as the Gibbs free energy difference from the ground-state structure to the more stable transition state. The rate constants for enantiomerization ($k_{ent}$) and racemization ($k_{rac}$), and half-life for racemization ($t_{1/2}$) were calculated.

**Racemization studies of the products**. The axially chiral products **9a**, **9n**, and **9o** (0.05 mmol) were dissolved in toluene (100 mL) separately and heated at different temperatures for 24 h. The ee value was tested every 1 h via HPLC on the chiral stationary phase.

## Data availability
The X-ray crystallographic coordinates for structures generated in this study have been deposited in the Cambridge Crystallographic Data Centre under accession code CCDC 2033533 [http://www.ccdc.cam.ac.uk/data_request/cif]. These data can be obtained free of charge. The density functional theory (DFT) calculation data generated in this study are provided as Source Data. All other data are available from the authors upon request. Source data are provided with this paper.

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

## Acknowledgements

We thank Dr. Yongxin Li (NTU) for their assistance with X-ray structure analysis. We acknowledge DFT calculation supports from Shenzhen HUASUAN Technology Co., Ltd. We acknowledge financial supports from Singapore National Research Foundation under its NRF Investigatorship (NRF-NRFI2016-06, Y.R.C.) and Competitive Research Program (NRF-CRP22-2019-0002, Y.R.C.); the Ministry of Education, Singapore, under its MOE AcRF Tier 1 Award (RG7/20, Y.R.C.; RG5/19, Y.R.C.), MOE AcRF Tier 2 (MOE2019-T2-2-117, Y.R.C.), MOE AcRF Tier 3 Award (MOE2018-T3-1-003, Y.R.C.); Nanyang Research Award Grant (Y.R.C.), Chair Professorship Grant (Y.R.C.), Nanyang Technological University; the National Natural Science Foundation of China (21772029, Y.R.C.; 21801051, Z.J.; 21961006, Z.J.; 22071036, Y.R.C.; 32172459, Z.J.; 81360589, W.T.); The 10 Talent Plan (Shicengci) of Guizhou Province ([2016]5649, Y.R.C.), The Science and Technology Department of Guizhou Province ([2019]1020, Y.R.C.; Qiankehejichu-ZK[2021]Key033, Z.J.), the Program of Introducing Talents of Discipline to Universities of China (111 Program, D20023, Y.R.C. and Z.J.) at Guizhou University; Frontiers Science Center for Asymmetric Synthesis and Medicinal Molecules, Department of Education, Guizhou Province [Qianjiaohe KY number (2020)004, Y.R.C.]; the Guizhou Province First-Class Disciplines Project [(Yiliu Xueke Jianshe Xiangmu)-GNYL(2017) 008, W.T.], Guizhou University of Traditional Chinese Medicine, and Guizhou University.

## Author contributions

J.Y. contributed to the reaction design and main experiments. R.M. contributed to the design and part of the experiments. S.R., J.X., and B.M. contributed to part of the experiments. T.L. and Z.J. contributed to mechanistic discussions and coordinated DFT calculations and data analysis that was performed by HUASUAN. W.T. and Y.R.C. conceptualized and supervised the research. J.Y. drafted early versions of the manuscript, with final revisions by Y.R.C. All authors contributed to discussions.

## Competing interests

The authors declare no competing interests.
