## [Peer Review File · Nature Communications]

REVIEWER COMMENTS

Reviewer #1 (Remarks to the Author):

Axially chiral alkenes are difficult to control the enantioselectivity due to low rotational barrier of the axis. In this manuscript, Chi and coworkers described an NHC-catalyzed atroposelective synthesis of the challenging axially chiral alkenes involving direct nucleophilic addition of sulfinic anion to active alkyne under mild reaction conditions with excellent stereoselectivities (up to >99:1 er, >20:1 E/Z) and good yields. Key steps involve selective acetylenic acylazolium intermediate and sequential E-selective protonation to construct the chiral axis. The sulfone and carboxylic ester moieties are common moieties in bioactive and other functional molecules and can be readily transformed to diverse axially chiral compounds. In addition, the heteroaryl substituents, such as quinolone, indole and benzothiophene were also suitable for this transformation, demonstrating that the substrate scope is more compatible than the previous methodology reported. It is a good complementary development in the field of construction of axially chiral compounds and should be of great interest to the readership of this journal. The procedures are clearly explained, products are fully characterized. On this basis, this manuscript deserves to be accepted for publication in Nature Communications after the following points addressed.

1. As regarding to substrate involved, the ether (OMe) is essential for good results. What is the main reason? Is there any interaction with catalyst? The author should explain this issue in the revised manuscript.
2. For most of the product, the E/Z selectivity was excellent. However, the selectivity of heteroaryl substituents was poor (products 9ad, 9ae and 9af). Any explanation is welcomed for a better understanding of the mechanism of the reaction.
3. As for the transformation from 9a to 10b, the enantioselectivity excess drops a lot. Could the authors give any comments or double-check the result?

Reviewer #2 (Remarks to the Author):

In this manuscript, an NHC-catalyzed stereoselective 1,4-addition to ynals to provide axially chiral styrenes is reported. The strategy is based on the amine-catalyzed, stereoselective 1,4-addition to ynals to give axially chiral styrenes as described in Nat. Commun. 8, 15238 (2017) – Tan, reference 33 – and the related synthesis of axially chiral sulfone-containing styrenes disclosed in J. Am. Chem. Soc. 140, 7056 (2018) – Yan, reference 34 –

High selectivities and yields were obtained after an optimization and a broad substrate scope was achieved when using 5 mol% of catalyst and 2 equiv. DQ, if the reaction was performed at -20°C over 6 to 9 days. The rotational barriers were determined to confirm DFT studies. Several transformations of a product prepared in 1.2 g show the utility of the products. The conclusions indicate that other nucleophiles were tested and that the results are encouraging.

Overall, the work in this manuscript is interesting and well-described. The method is simple and can be used by others without obvious difficulty. The conception advance is reasonable and it is thus recommended to accept this manuscript after minor revisions as listed below.

- 1: Title ...Bearing Acyclic Alkenes... repeats the alkene included in styrene and is therefore unnecessary.
- 2: Figure 1b, there is a problem in the formatting of ...Brønsted...
- 3: Figure 1b, it is important to mention that this nucleophile was already used in – reference 34 – and it would therefore be appropriate to list the added nucleophiles that are already reported.
- 4: Figure 1c, it would be instructive to also indicate the oxidation step in the mechanism.
5. Figure 2a and Figure 2b are rather technical and should be moved to the SI.

Reviewer #3 (Remarks to the Author):

The manuscript by Tian, Chi, and coworkers presents an interesting strategy to prepare axially chiral styrenes. The authors have used experiments and DFT calculations to comment on the stability of the chiral products. Some minor points that need to be addressed are as follows:

- 1) In the SI, two TSs, TS9a and TS9a' are mentioned. It is stated that the rotational barrier with respect to the more stable TS is reported. This is a bit confusing as TS9a' has the lower barrier. Also, the nature of the TS9a', which lies very close to 9a, is not clear. The authors can clarify this part.
- 2) It would be helpful if the authors can mention the magnitude of the vibrational frequencies obtained for the TS. Rotational TSs can have low frequencies, and hence, it becomes difficult to identify the true nature. In addition, IRC calculations should be carried out to verify the nature of both TSs.

Garima Jindal

Point-by-point response to the reviewers' comments
(Manuscript ID: NCOMMS-21-28650)

Reviewer #1 pointed out that “On this basis, this manuscript deserves to be accepted for publication in *Nature Communications* after the following points addressed.”

- 1) **Reviewer's comment:** As regarding to substrate involved, the ether (OMe) is essential for good results. What is the main reason? Is there any interaction with catalyst? The author should explain this issue in the revised manuscript.

Our Response: We thank the reviewer for pointing out such important characteristic of our reaction. To verify whether there is any special function of the 2-alkoxy substituents (e.g., OMe), we performed an additional reaction with 2-ethyl substituted substrate and found a remarkable decrease in both *e.r.* and *E/Z* values. Inspired by the coordination effect of Lewis acid with nucleophiles and 2-methoxyl substituted acetylenic acylazolium intermediates [Wang, J. et al. *Nat. Commun.* **9**, 611 (2018)], we hypothesized that there might be hydrogen bond interactions between nucleophiles and 2-alkoxy substituents in our catalytic system, which probably had a positive effect on the enantio- and *E/Z* selectivity (Fig. R-1). We have included this additional example (**9m**) and explanation in the revised manuscript and Supplementary Information (Supplementary Fig. 190).

Figure R-1 Additional example and plausible hydrogen bond interaction

Besides, the presence of 2-alkoxy substituents (e.g., OMe) might have other functions which benefit the reaction. For example, the high active acetylenic acylazolium intermediate (**II**) may resonate to intermediate **II'** which may react with sulfinate anion to provide allenolate **III** (Fig. R-2). The allenolate **III** then undergoes protonation and esterification to afford the product. We have also included this possibility in the Supplementary Information (Supplementary Fig. 191).

Figure R-2 Plausible resonance intermediates involved pathway

We don't know whether there is any interaction between the ether groups (e.g., OMe) with our catalyst. Because we don't have any evidence to support or rule out this possibility.

- 2) **Reviewer's comment:** For most of the product, the *E/Z* selectivity was excellent. However, the selectivity of heteroaryl substituents was poor (products **9ad**, **9ae** and **9af**). Any explanation is welcomed for a better understanding of the mechanism of the reaction.

Our Response: The poor *E/Z* selectivity for **9ae-9ag** (previously numbered as **9ad-9af**) might attribute to the less steric hindrance of the (hetero)aryl rings on the ynal skeletons. As shown in Fig. R-1, the *E/Z* selectivity of our products comes from the protonation of allenolate intermediate (**III**). For most of the products, the naphthalene rings on the ynal substrates provided enough steric hindrance to prevent the rotation of allenolate intermediate (**III**). However, for **9ae-9ag** (previously numbered as **9ad-9af**), the less steric hindrance of aromatic rings on the ynals led to more flexibility of the allenolate intermediates (**III**), which may rotate from **III** to **III'** (Fig. R-3). The hydrogen bond interaction modes for intermediates **III** and **III'** were different, and intermediate **III'** would prefer to perform the *Z*-selective protonation, which then resulted in the non-axially chiral product with *Z*-alkene. We have explained this issue in the manuscript and Supplementary Fig. 192.

Figure R-3 Plausible reasons for the low *E/Z* selectivity of **9ae-9ag**

- 3) **Reviewer's comment:** As for the transformation from **9a** to **10b**, the enantioselectivity excess drops a lot. Could the authors give any comments or double-check the result?

Our Response: We have double-checked this reaction and got the same result. During the reaction, we had to warm up the reaction mixture to 0 °C in order to promote the Grignard addition process since ^tPrMgBr is very bulky and the addition reaction was too slow under low temperature. Probably some reaction intermediate in the process was not so stable, which resulted in the decrease of *ee* value.

Reviewer #2 pointed out that “The conception advance is reasonable and it is thus recommended to accept this manuscript after minor revisions as listed below.”

- 1) **Reviewer's comment:** Title ...Bearing Acyclic Alkenes... repeats the alkene included in styrene and is therefore unnecessary.

Our Response: We have revised the title and deleted the “Bearing Acyclic Alkenes”.

- 2) **Reviewer's comment:** Figure 1b, there is a problem in the formatting of ...Brønsted...

Our Response: We have corrected this mistake.

- 3) **Reviewer's comment:** Figure 1b, it is important to mention that this nucleophile was already used in – reference 34 – and it would therefore be appropriate to list the added nucleophiles that are already reported

Our Response: We have included the nucleophiles which have already been reported, and listed them in the revised manuscript.

- 4) **Reviewer's comment:** Figure 1c, it would be instructive to also indicate the oxidation step in the mechanism.

Our Response: We have added the oxidation step in Figure 1c.

- 5) Reviewer's comment: *Figure 2a and Figure 2b are rather technical and should be moved to the SI.*

Our Response: We have moved the corresponding information (Figure 2a and Figure 2b) to the Supplementary Information.

Reviewer #3 pointed out that “*The manuscript by Tian, Chi, and coworkers presents an interesting strategy to prepare axially chiral styrenes. The authors have used experiments and DFT calculations to comment on the stability of the chiral products. Some minor points that need to be addressed are as follows:*”

- 1) Reviewer's comment: *In the SI, two TSs, **TS9a** and **TS9a'** are mentioned. It is stated that the rotational barrier with respect to the more stable TS is reported. This is a bit confusing as **TS9a'** has the lower barrier. Also, the nature of the **TS9a'**, which lies very close to **9a**, is not clear. The authors can clarify this part.*

Our Response: Sorry for the mistake. **TS9a'** is actually a stable state (the enantiomer of **9a**) rather than a transition state. Because **TS9a'** has already been rotated for about 180° in respect to **9a**. That is why the nature of the **TS9a'** lies very close to **9a**. Thus we have changed the numbering of this state from **TS9a'** to *ent-9a*.

- 2) Reviewer's comment: *It would be helpful if the authors can mention the magnitude of the vibrational frequencies obtained for the TS. Rotational TSs can have low frequencies, and hence, it becomes difficult to identify the true nature. In addition, IRC calculations should be carried out to verify the nature of both TSs.*

Our Response: According to the suggestion, we have provided the magnitude of the vibrational frequency and IRC calculation result for transition state **TS9a** in the Supplementary Information. In contrast, the data for *ent-9a* (previously numbered as **TS9a'**) is not provide since *ent-9a* is not a transition state and has no vibrational frequency and IRC calculation data.

REVIEWERS' COMMENTS

Reviewer #1 (Remarks to the Author):

The authors have addressed most of the points raised. Therefore, the current manuscript deserves to be accepted for publication in Nature Communications without any delay.

Reviewer #3 (Remarks to the Author):

The authors have addressed the comments, and the manuscript can be accepted.

October 26, 2021

Point-by-point response to the reviewers' comments
(Manuscript ID: NCOMMS-21-28650A)

Reviewer #1: *The authors have addressed most of the points raised. Therefore, the current manuscript deserves to be accepted for publication in Nature Communications without any delay.*

Our Response: Thanks very much.

Reviewer #3: *The authors have addressed the comments, and the manuscript can be accepted.*

Our Response: Thanks very much.